# Designing a deposit-refund system for cigarette butts: What do smokers care about?

**Fivita Stri[1], Takuro Uehara[1]\*, Takahiro Tsuge[2], Sitadhira Prima Citta[3], Misuzu Asari[4]**

**1** College of Policy Science, Ritsumeikan University, Osaka, Ibaraki, Japan, **2** Graduate School of Global Environmental Studies, Sophia University, Chiyoda-Ku, Tokyo, Japan, **3** Research Organization of Open Innovation and Collaboration, Ritsumeikan University, Osaka, Ibaraki, Japan, **4** Research Institute for Humanity and Nature, Kita-Ku, Kyoto, Japan

\* takuro@fc.ritsumei.ac.jp

## Abstract

Cigarette butts (CBs) are the world's most littered item and significantly contribute to environmental pollution. A deposit-refund system (DRS) has been proposed to reduce CB littering, but its effective design remains underexplored. This study addressed this gap by investigating smokers' perceptions and preferences in hypothetical DRS scenarios for CBs. We conducted a discrete choice experiment in Japan (n = 1,865) and Indonesia (n = 2,000). Respondents were divided into treatment and control groups, with the treatment group receiving information on CB environmental impact aligned with the WHO's campaign. Our results revealed that a DRS for CBs was preferred to the status quo, with higher preferences in Indonesia (90.33%) than in Japan (63.92%). The information treatment further increased DRS preferences (Indonesia: 91.82%; Japan: 69.83%) and willingness to endure cost to support DRS operations. Cost simulations showed participation probabilities remained above 55% with a cost of up to 5% of a cigarette's price in Japan, and above 80% with a cost of up to 10% in Indonesia. Our findings underscore the importance of environmental information in DRS adoption. Both countries preferred a producer-managed system to a government-managed one, highlighting an opportunity for tobacco producers to fulfill extended producer responsibility through a DRS. Furthermore, DRS design should be country-specific. Notably, Japanese respondents' familiarity with heat-not-burn cigarettes influenced their preference for a tailored DRS to those products, whereas Indonesian respondents preferred a DRS for CBs. Japanese respondents also emphasized accessibility more than their Indonesian counterparts.

## 1. Introduction

Littering is a major contributor to environmental pollution [1], with cigarette butts (CBs) being the most littered item, accounting for 4.5 trillion discarded worldwide yearly [2–4]. Their ubiquity is evident across diverse environments, from urban

**Data availability statement:** All relevant data are within the manuscript and its Supporting information files.

**Funding:** TU and MA were funded by the Environment Research and Technology Development Fund of the Environmental Restoration and Conservation Agency of Japan [grant number JPMEERF21S11920]. URL: https://www.erca.go.jp/erca/english/index.html The funder did not play any role in the study design, data collection and analysis, decision to publish, or preparation of the manuscript.

**Competing interests:** The authors have declared that no competing interests exist.

streets and parks [5–7] to beaches and marine ecosystems, where more than 1.8 million CBs were retrieved during ocean clean-up [8]. The environmental impact of CBs is compounded by their durability, chemical toxicity, and leaching potential [9]. CBs, typically composed of cellulose acetate filters, are not biodegradable under natural conditions and may persist in the environment for up to 14 years [10–12]. Each discarded CB can leach a complex mixture of over 7,000 harmful chemicals, including nicotine, heavy metals (such as cadmium, lead, and arsenic), polycyclic aromatic hydrocarbons, polycyclic aromatic compounds, and benzene [12–15]. These contaminants have been shown to adversely affect soil quality, inhibit plant growth, and pose ingestion risks to aquatic organisms [16–21].

Reducing CB littering has become a global policy focus. Measures, such as smoke-free regulations, taxes, fines, and clean-up initiatives, have been implemented; however, CB littering remains pervasive [7,13,16]. The WHO Framework Convention on Tobacco Control and European Union Directives proposed banning cigarette filters and developing sustainable alternatives [3,22]. However, some studies suggested that biodegradable filters may be unmarketable and may even encourage littering [23,24]. Challenges stem from CBs' small size, low economic value, and ingrained disposal habits [9,13,25]. Tossing CBs is often a habitual part of smoking, and is widely considered acceptable [13]. Behavioral nudges such as voting ashtrays have been proposed [26]; however, recent efforts are shifting toward mitigating the environmental impact of CBs [3,22].

Recently, a deposit-refund system (DRS) has been proposed as a promising solution to address CB littering [9,12,27,28]. Under this system, smokers pay a deposit included in the cigarette price, refunded upon returning CBs. Two key features make the DRS promising: financial incentives and support for recycling or upcycling. First, the refund offers financial incentives to motivate returns [9], reduces litter and cleanup costs, and creates income opportunities for informal waste collectors. Second, the DRS can support CB recycling or upcycling, currently incinerated or landfilled, both of which are unsuitable [25], into products such as bricks, roads, energy sources, or insecticides [12,29]. Therefore, the DRS addresses current waste management gaps by offering safe disposal options [25]. Following high return rates for other recyclables, the DRS shows similar promise for CBs [9]. Implementation is feasible, as demonstrated in Japan, where collection systems for heat-not-burn (HNB) cigarette components have been established. HNB cigarettes are electronic tobacco products in which the tobacco is heated, rather than combusted, to generate an aerosol. The collection systems cover components such as devices, batteries, cartridges, and capsules, owing to fire safety considerations [30–32]. Technological integration is also viable; for example, radio frequency identification (RFID) tags can be embedded in filters for hands-free, trackable collection. While current RFID costs are approximately 5 JPY per unit [33], prices are expected to decline.

Despite earlier studies highlighting the potential of a DRS for CBs, two research gaps remain. First, smokers' perceptions and preferences regarding a DRS for CBs remain underexplored. Adopting new waste collection systems depends on consumer preferences and expected behavior [34]. While Hoek et al. [35] found that 51% of smokers

preferred a fixed 2 USD deposit, they did not explore how preferences might vary across different deposit levels. Additionally, although they examined how information about the environmental impact of CB affects perceived responsibility, its influence on the perceptions of a DRS and preferences remains underexplored. Providing such information may increase support for a DRS, aligning with the WHO campaigns to raise awareness of tobacco's environmental impact [3,4]. Many smokers believe that CBs are biodegradable, a misconception linked to higher littering rates, which also underscores the need for targeted information interventions [36–38]. This highlights the need for further in-depth research using hypothetical scenarios, enabling the assessment of different attributes, levels, and environmental information to better understand preferences and perceptions that can inform the development of an effective DRS design. Second, studies assessing a DRS for CBs across different socio-economic contexts, particularly between developed and developing countries, are lacking. Developing countries, which often experience higher rates of CB littering, have received limited attention regarding targeted measures [1].

To address these gaps, this study investigated smokers' perceptions of, and preferences for a DRS for CBs, aiming to propose an effective DRS design. We conducted a comparative study using a treatment-control design in Japan and Indonesia, two countries selected for their potential to yield generalizable insights [39]. Both countries face CB littering issues [40–42], and differ in smoker characteristics such as age and gender [43–47]. The following research questions were explored:

RQ1. What do smokers understand about the environmental impact of CBs?

RQ2. How do smokers perceive a DRS for CBs?

RQ3. What are smokers' preferences for a DRS?

RQ4. How does information on the environmental impact of CBs affect smokers' preferences for a DRS?

RQ5. What smokers' characteristics influence their preferences for a DRS for CBs over the status quo?

## 2. Materials and methods

This study applied a discrete choice experiment (DCE) and a treatment-control design to address research questions 3, 4, and 5. This study was approved by the affiliated university's ethics committee (No. Kinugasa-hito-2024-21).

### 2.1 Experimental design

Because a DRS for CBs has not been implemented or tested, understanding smokers' perceptions and preferences in a hypothetical setting is crucial for its design. We used a DCE, which elicits stated preferences by asking respondents to choose the option that provides the highest utility from a set of hypothetical alternatives [48,49]. DCE has been applied to DRS [50,51]. In this study, respondents evaluated two mandatory DRS per choice set, both requiring a deposit, with the ability to opt-out for preference accuracy [52,53]. Using an unlabeled DCE (i.e., DRS 1 and DRS 2), we analyzed the trade-offs between the attributes and their levels [48,54]. From the 48 possible choice sets (4 × 2 × 2 × 3, Table 1), we reduced to 12 by adopting Lorenzen and Anderson's [55] orthogonal main effects design.

We used a treatment-control design to evaluate the impact of information intervention by comparing a treatment group that received the intervention, with a control group that did not [56]. The treatment groups received environmental information on the impact of CBs, and completed a comprehension test to assess their understanding of this information. Respondents who failed the test twice were excluded from analysis.

### 2.2 Questionnaire design

The questionnaire comprised six components: (1) sociodemographic characteristics, (2) behavior, attitude, and awareness, (3) information intervention and comprehension test for treatment groups, (4) DCE choice sets, (5) perception of

**Table 1. DCE attributes and levels.**

| Attribute | Level | | Information sources |
|---|---|---|---|
| | **Japan** | **Indonesia** | |
| Deposit | 5 JPY/pack<br>50 JPY/pack<br>100 JPY/pack<br>200 JPY/pack | 300 IDR/pack<br>1,500 IDR/pack<br>3,000 IDR/pack<br>6,000 IDR/pack | [35,51,68–73], FGs |
| Refund rate | 100%<br>50% | | [59,69], FGs |
| Management institution | Government<br>Producer | | [3,22,23,74–76], FGs |
| Accessibility | 5 minutes<br>10 minutes<br>30 minutes | | [34,50,59,77,78], FGs |

JPY, Japanese Yen; IDR, Indonesian Rupiah. 1 JPY = 0.0068 USD, 1 IDR = 0.000062 (August 24, 2025, from https://www.google.com/finance/).

DRS, and (6) HNB cigarettes. The control group questionnaire excluded component (3). S2 Supplementary provides a sample questionnaire.

Before finalizing the questionnaire, a focus group (FG) was conducted in each country to validate DCE attributes and levels, assessing comprehensibility, relevance, distinctiveness of levels, reasonableness, and preference [57]. The FGs also evaluated the environmental information's clarity, novelty, and significance. Japanese participants were probability sampled via the survey company iBRIDGE (https://freeasy24.research-plus.net/) between September 5 and 6, 2024, whereas Indonesian participants were recruited via snowball sampling between August 10 and 21, 2024. Both FGs were conducted online and recorded on September 15 and 17, 2024, for Indonesia and Japan, respectively (S1 Supplementary provides the FGs results). Online informed consent was obtained from all participants prior to their involvement in the FGs. Participants were required to explicitly agree to the terms outlined in the consent form before proceeding with the sessions. The consent process was documented electronically, with records retained for verification. Two pre-tests were also conducted in Japan via the survey company iBRIDGE to check the questionnaire's logic: the first on October 23–24, 2024 (n = 306), and the second on October 29–30, 2024 (n = 260). Informed consent was obtained from all participants electronically at the beginning of the survey. Participants were required to acknowledge their consent before proceeding with the survey.

Component 1 included sociodemographic characteristics such as gender, age, education, and annual household income to reveal their relationship with DRS preferences. Although previous studies showed that these characteristics significantly influence compliance with DRS in other waste contexts, the findings remain inconclusive [34,50,58].

Component 2 assessed behavior, attitude, and awareness to understand smokers' characteristics and their relationship with DRS preferences. First, behavior was captured by daily cigarette consumption and littering frequency. Both influence return in DRS [59]. To address social desirability bias when asking about illegal behaviors such as littering [60], we applied inferred valuation by asking respondents to estimate how frequently other smokers litter CBs. Inferred valuation could more accurately reflect actual behavior than self-reports, evident in the context of illegal eel consumption [61]. Second, attitude was captured by eco-guilt, a widely studied factor in pro-environmental behavior [62–65]. Eco-guilt refers to individual guilt upon failing to meet personal or societal environmental standards [66]. Eco-guilt could heighten smokers' sense of responsibility for littering CBs, and motivate them to choose a DRS over opting out to align with environmental standards. We adopted five eco-guilt items with the highest mean loadings from Ágoston et al. [62] (S1 Appendix). Three to five items are recommended for Cronbach's alpha [67]. Third, awareness was captured by knowledge of the environmental impact of CBs, revealing both smokers' understanding (RQ1), and its relationship with DRS preferences. We adopted four items from Hoek et al. [35].

In component 3, the treatment groups received the environmental information intervention before answering the DCE questions, which were presented in each country's respective language (i.e., Japanese and Indonesian) (Fig 1). Three blanks within the information were used in the comprehension test (Fig 2).

In component 4, each choice set in our DCE presented two DRS and an opt-out option featuring four attributes: deposit, refund rate, management institution, and accessibility (Fig 3). The attributes and levels were selected based on previous studies and validated through FGs (Table 1). Before the choice sets, participants were provided with a scenario explaining the DRS, including its operation, assumptions about cigarette and pack pricing, attribute definitions, CB recycling, and the option to return any number of CBs (S2 Appendix).

The first attribute, deposit, refers to an additional fee paid when purchasing cigarettes, that is refundable (fully or partially, depending on the assumed refund rate) upon returning CBs. Based on FGs feedback, we set the deposit ratios

# CIGARETTE BUTTS ENVIRONMENTAL IMPACT

- **Non-biodegradable***
- Toxic: nicotine, heavy metals, and **7000** other chemicals
- Micro(nano)plastic fibers

*Non-biodegradable= not capable of being broken down by the action of living organisms

(Akhbarizadeh et al., 2021; Marinello et al., 2020)

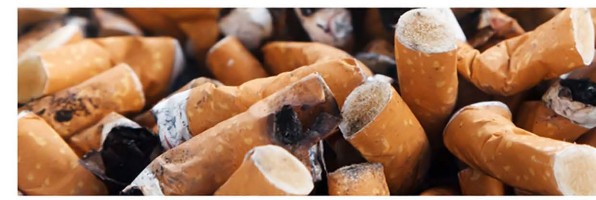

**4.5 trillion** cigarette butts are littered each year, globally
(WHO, 2017)

Reduce plants' growth by more than **25%**
(Green et al., 2019)

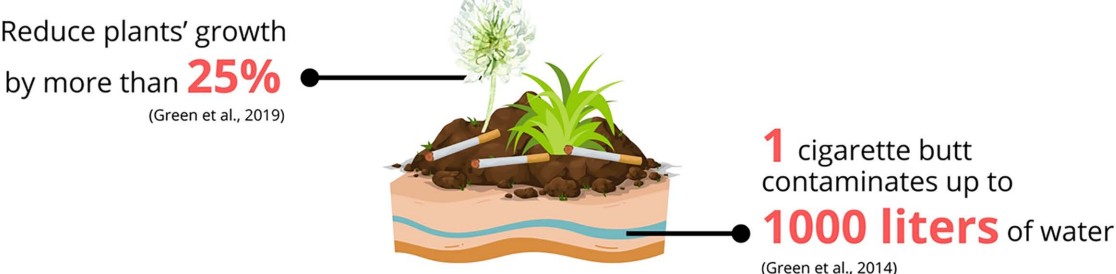

**1** cigarette butt contaminates up to **1000 liters** of water
(Green et al., 2014)

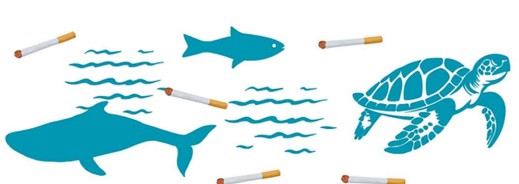

- **No.1** items collected during global ocean cleanup. Surpassed plastic bottles and food wrappers
(Ocean conservancy, 2023)

- Poison and kill marine animals
(Dobaradaran et al., 2021)

**Fig 1. Environmental information intervention for treatment groups.**

# CIGARETTE BUTTS ENVIRONMENTAL IMPACT

- **(1)**
- Toxic: nicotine, heavy metals, and **7000** other chemicals
- Micro(nano)plastic fibers

*Non-biodegradable= not capable of being broken down by the action of living organisms

(Akhbarizadeh et al., 2021; Marinello et al., 2020)

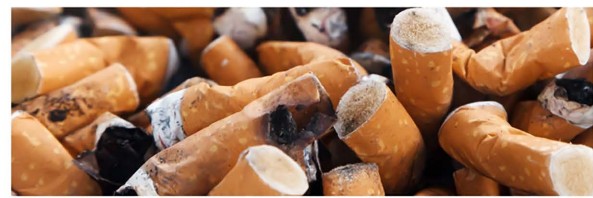

**4.5 trillion** cigarette butts are littered each year, globally
(WHO, 2017)

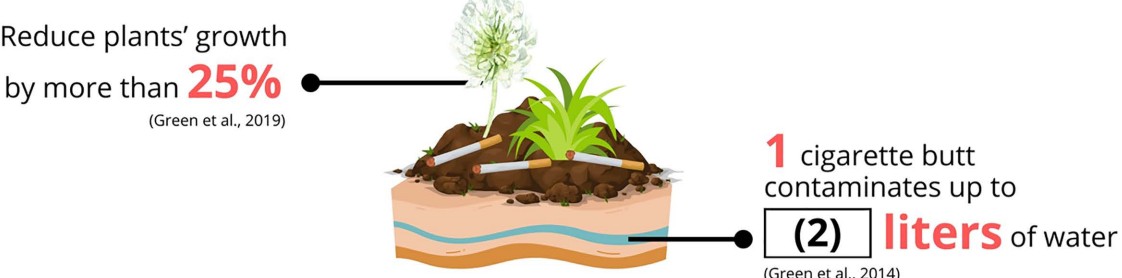

Reduce plants' growth by more than **25%**
(Green et al., 2019)

**1** cigarette butt contaminates up to **(2) liters** of water
(Green et al., 2014)

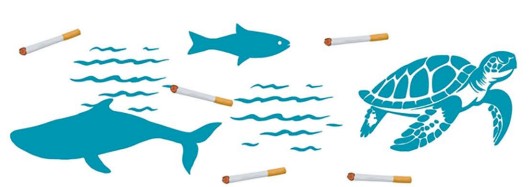

- **No.** **(3)** items collected during global ocean cleanup. Surpassed plastic bottles and food wrappers
(Ocean conservancy, 2023)

- Poison and kill marine animals
(Dobaradaran et al., 2021)

**Fig 2. Illustration used to test the comprehension of environmental information intervention for treatment groups.**

at 1%, 10%, 20%, and 40% of the average cigarette pack price in Japan (500 JPY for 20 sticks), and 1%, 5%, 10%, and 20% in Indonesia (32,000 IDR for 16 sticks). These ratios were also cross-referenced from previous studies on DRS for CBs [35], and for beverage packaging [68,72,73]. The second attribute, refund rate, refers to the percentage of the deposit refunded. We included a refund rate of 50% to understand smokers' preferences regarding their contribution to the DRS and the environment. In the scenario, we explained that the remaining amount would be used to cover the operating costs of the DRS under the assumed managing institution. The third attribute, management institution, refers to the entity operating the DRS. While governments have traditionally managed CBs [23], the EU Directive [22] and the WHO [3] have promoted Extended Producer Responsibility (EPR). Under EPR, a producer-managed DRS can help mitigate external-ities from CB littering, such as environmental pollution [76]. The fourth attribute, accessibility, reflects the effort required

Combination 1. There are two deposit-refund systems below. Please choose the system you prefer.

| Combination 1 | Deposit-refund system 1 | Deposit-refund system 2 |
|---|---|---|
| Deposit | 100 JPY/pack | 100 JPY/pack |
| Refund rate | 100% | 50% |
| Management institution | Government | Government |
| Accessibility | 5 minutes | 5 minutes |

a. I prefer the deposit-refund system A

b. I prefer the deposit-refund system B

c. I prefer neither of them

**Fig 3. An example of a DCE question (Japan, Block 1, Question 1).**

for returns, such as time or burden, and perceived convenience [34,50,59]. Accessibility was evaluated by the estimated travel time to return points, which is a common proxy in consumer behavior research [77,78].

Component 5 encompassed an open-ended question to explore smokers' perceptions of DRS for CBs through free association (RQ2). We placed this component after DCE choice sets to minimize misinformation regarding DRS, which could influence respondents' perspectives [34,79].

Component 6 examined smokers' usage of, familiarity with, and preferences between conventional and HNB cigarettes in the DRS context. HNB cigarettes were prioritized over other electronic tobacco products because they resemble CBs regarding disposal. HNB cigarette use is increasing in both Japan and Indonesia [80–82]. In Japan, the tobacco company has implemented nationwide collection programs through retailers, allowing the return of devices, batteries, cartridges, and capsules owing to the fire risks associated with HNB products [30–32]. However, this program excludes non-combusted cigarette sticks used in HNB products. Therefore, understanding which tobacco products smokers prefer is crucial for promoting the adoption of a DRS.

## 2.3 Data analysis

We applied conditional logit models to analyze smokers' DRS preferences obtained from DCE (RQ3 and RQ4). Following Aizaki et al.'s [48] random utility theory, the systematic component of utility for individual $n$ for alternative $i$ can be specified as:

$$V_{in} = ASC + \beta_1 Cost_{in} + \beta_2 Management\ institution_{in} + \beta_3 Accessibility_{in} \tag{1}$$

where ASC is an alternative-specific constant, and $\beta_1$, $\beta_2$, and $\beta_3$ are coefficients of the specified attributes. We considered using deposit and refund rate as separate attributes to compute conditional logit models, following previous studies that used deposit [51,83]. However, the results were inconclusive (see S3 Appendix). This suggests that respondents did not perceive these attributes independently; rather, they viewed them as a combined value, which we defined as cost (i.e., the product of deposit and refund rate). A 100% refund rate was assigned a cost of zero, while a 50% refund rate was assigned a cost equal to half the deposit. Respondents were informed of the remaining deposit in the DRS operation scenario. Therefore, the cost variable reflects the actual monetary burden of using DRS to support its operation. We used cost in other estimations as well. ASC was assigned a value of "1" for a DRS and "0" for opt-out. The categorical

attribute (management institution) was effect-coded to ensure that the base-level variable (government) was interpretable independently of ASC [84]. Accessibility ($\beta_3$) captures the value of time-consuming, or otherwise uncomfortable actions not reflected in the previously assigned zero cost under a 100% refund rate. To examine how smokers' characteristics influence DRS preferences (RQ5), the following equation included interaction terms between ASC and sociodemographic factors (gender, age, education, and annual household income), as well as behavioral, attitudinal, and awareness variables (cigarette consumption, littering frequency, inferred valuation of littering frequency, eco-guilt, and knowledge about CBs). The knowledge codes were reversed for certain items to make the results more intuitive:

$$
\begin{aligned}
V_{in} = {} & ASC + \beta_1 Cost_{in} + \beta_2 Management\ institution_{in} + \beta_3 Accessibility_{in} + \beta_4 (ASC \times Gender_n) \\
& + \beta_5 (ASC \times Age_n) + \beta_6 (ASC \times Education_n) + \beta_7 (ASC \times Income_n) \\
& + \beta_8 (ASC \times Cigarettes\ consumption_n) + \beta_9 (ASC \times Littering_n) \\
& + \beta_{10} (ASC \times Littering\ inferred_n) + \beta_{11} (ASC \times Eco\ guilt_n) + \beta_{12} (ASC \times Knowledge_n)
\end{aligned}
\tag{2}
$$

Based on Equation 1, we computed the willingness to pay (WTP) by dividing the coefficients of the non-monetary attributes by the coefficient of the monetary attribute [48]. The equation is given by:

$$
WTP = -\frac{\beta_n}{\beta_1}, \quad n \in \{2,\ 3\}
\tag{3}
$$

To make the WTP comparable across countries, we computed the percentage of WTP by dividing it by the price of a cigarette pack introduced in DCE choice sets (500 JPY for Japan and 32,000 IDR for Indonesia) [85]. All DCE analyses were conducted using the statistical software R (version 4.3.3) with the mded packages [48].

$$
WTP\ (\%) = \frac{WTP}{Reference\ price}
\tag{4}
$$

To address RQ2 on smokers' DRS perceptions, we conducted an exploratory qualitative analysis using inductive coding [86]. Codes were assigned based on themes that emerged directly from the data [87]. To ensure coverage of respondents' perceptions, multiple codes, including options such as "don't know," "don't understand," and "nothing in particular," were allowed [88]. The lead author first assigned the codes, and the two coauthors reviewed them through multiple rounds of inductive coding, with revisions made as necessary.

## 2.4 Data collection

We recruited 4,116 respondents using survey companies (n = 2,116 from Japan; n = 2,000 from Indonesia). The Japanese survey data were collected by iBRIDGE from November 22 to December 2, 2024, and the Indonesian survey data were collected by Koeeru (https://koeeru.com/) from December 27, 2024, to January 17, 2025. Smoking prevalence using conventional cigarettes, gender, and age were used in recruiting a diverse sample comprising active smokers. However, we restricted the ages to 20 years and above for Japan and 21 years and above for Indonesia, in accordance with the legal smoking age in each country. Informed consent was obtained from all participants electronically at the beginning of the survey. Participants were required to acknowledge their consent before proceeding with the survey.

## 3. Results

### 3.1 Sample description

A total of 3,865 valid responses were collected after excluding those who failed the comprehension test twice (n = 1,865 from Japan; n = 2,000 from Indonesia). Table 2 presents respondents' sociodemographic characteristics. Overall, the gender and age distributions of our sample represent the populations of both countries.

**Table 2. Sociodemographic characteristics of respondents (%).**

| Characteristic | Japan (n = 1,865) | | | | Indonesia (n = 2,000) | | | |
|---|---|---|---|---|---|---|---|---|
| | Scale | Control (n = 911) | Treatment (n = 954) | Pop dist. | Scale | Control (n = 1,000) | Treatment (n = 1,000) | Pop dist. |
| Gender | Male | 75.85 | 76.94 | 77.50 | Male | 95.30 | 95.90 | 97.00 |
| | Female | 24.15 | 23.06 | 22.50 | Female | 4.70 | 4.10 | 3.00 |
| Age | 20-29 | 10.54 | 10.69 | 11.80 | 21-29 | 27.10 | 23.40 | 24.34 |
| | 30-39 | 15.15 | 16.98 | 15.30 | 30-39 | 26.30 | 30.50 | 26.94 |
| | 40-49 | 26.89 | 26.52 | 22.00 | 40-49 | 23.40 | 21.00 | 23.08 |
| | 50-69 | 39.96 | 40.67 | 38.10 | 50-69 | 23.20 | 25.10 | 25.65 |
| | Above 70 | 7.46 | 5.14 | 12.90 | | | | |
| Education | Middle school | 3.40 | 2.94 | – | Did not complete elementary school | 0.30 | 0.40 | – |
| | High school | 31.83 | 30.40 | – | Elementary school or equivalent | 0.80 | 0.40 | – |
| | Vocational school | 13.28 | 13.21 | – | Middle school or equivalent | 2.40 | 2.10 | – |
| | University undergraduate | 47.64 | 48.74 | – | High school, vocational school, or equivalent | 39.10 | 32.00 | – |
| | University graduate | 3.84 | 4.72 | – | Diploma | 11.50 | 10.90 | – |
| | | | | | University undergraduate | 43.30 | 52.00 | – |
| | | | | | University graduate | 2.40 | 1.60 | – |
| | | | | | Doctorate | 0.20 | 0.60 | – |
| Annual household income in million (JPY/IDR) | Below 1 | 4.94 | 5.87 | – | Below 50 | 32.00 | 25.70 | – |
| | 1 to below 2 | 6.81 | 7.02 | – | 50 to below 60 | 11.40 | 11.00 | – |
| | 2 to below 3 | 9.99 | 10.80 | – | 60 to below 70 | 7.60 | 7.30 | – |
| | 3 to below 4 | 11.53 | 11.64 | – | 70 to below 80 | 5.90 | 6.00 | – |
| | 4 to below 5 | 12.84 | 10.48 | – | 80 to below 90 | 6.10 | 4.90 | – |
| | 5 to below 6 | 11.75 | 10.38 | – | 90 to below 100 | 5.00 | 5.40 | – |
| | 6 to below 7 | 9.22 | 6.50 | – | 100 to below 110 | 6.60 | 7.80 | – |
| | 7 to below 8 | 8.56 | 9.33 | – | 110 to below 120 | 6.80 | 8.10 | |
| | 8 to below 9 | 6.26 | 6.18 | – | Above 120 | 18.60 | 23.80 | |
| | 9 to below 10 | 5.71 | 7.55 | – | | | | |
| | 10 to below 12 | 6.48 | 6.08 | – | | | | |
| | 12 to below 15 | 2.85 | 4.72 | – | | | | |
| | 15 to below 18 | 0.88 | 1.05 | – | | | | |
| | 18 to below 20 | 0.66 | 0.84 | – | | | | |
| | Above 20 | 1.54 | 1.57 | – | | | | |

Population data were calculated using the Portal Site of Official Statistics of Japan and Statista and Statistics Indonesia and the Global Adult Tobacco Survey for Japan and Indonesia, respectively. However, national data on the distribution of educational levels and annual household income were unavailable.

The behavioral and attitude characteristics differed significantly between Japanese and Indonesian respondents, as shown in Fig 4. While Japanese respondents reported a higher mean daily cigarette consumption, their mean littering frequencies were lower than those of Indonesian respondents, both in self-reported frequencies and inferred valuations

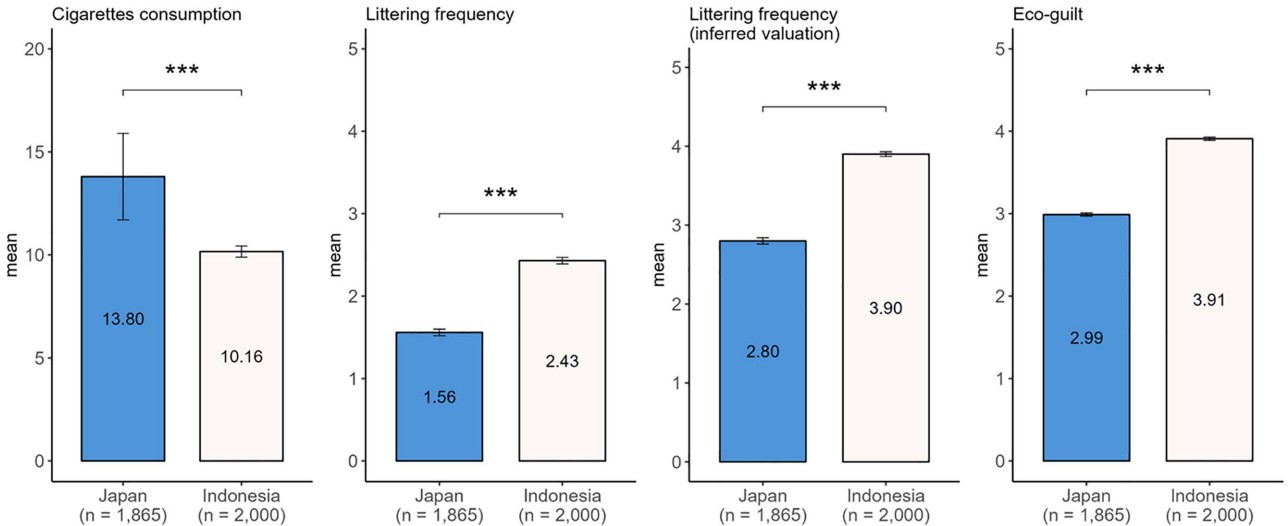

**Fig 4. Behavior and attitude characteristics of the respondents.** Welch's t-test. ***p < 0.001; **p < 0.01; *p < 0.05. Error bars indicate 95% confidence intervals. S4 Appendix provides the complete t-test results.

of other smokers' frequencies. The inferred values were higher in both countries. The eco-guilt scale showed high internal consistency, with Cronbach's alpha values of 0.89 (Japan) and 0.816 (Indonesia), both within the acceptable range of 0.70 to 0.90 [89]. Indonesian respondents exhibited a higher mean eco-guilt than their Japanese counterparts.

### 3.2 Understanding the environmental impact of CBs

Table 3 presents respondents' understanding of the environmental impact of CBs (RQ1). Overall, Japanese and Indonesian respondents exhibited similar levels of comprehension, with the lowest correct responses related to biodegradability (38.55% and 32.40%, respectively), and the highest related to toxicity (73.24% and 75.75%, respectively).

### 3.3 Perception of DRS

Fig 5 illustrates the results of the inductive coding of respondents' thoughts, ideas, or word associations toward a DRS for CBs (RQ2). Nine themes were identified. Compared with the majority of Indonesian respondents, where 43.90% and 42.80% in the control and treatment groups, respectively supported a DRS, most Japanese respondents opposed DRS ("Not support"), with 25.14% in the control group; however, this decreased to 18.24% in the treatment group. Environment-related responses were more prevalent among Indonesian respondents, even in the control group (30.40% versus 7.79% in Japan). Expectedly, environment-related opinions increased in the treatment groups in both countries

**Table 3. Comprehension of the environmental impact of CBs.**

| Statement | Japan (n = 1,865) | | | Indonesia (n = 2,000) | | |
|---|---|---|---|---|---|---|
| | Strongly agree or agree | Strongly disagree or disagree | Neither | Strongly agree or agree | Strongly disagree or disagree | Neither |
| Cigarette butts are biodegradable | 21.77% | **38.55%** | 39.68% | 39.25% | **32.40%** | 28.35% |
| Cigarette butts are toxic to the environment | **73.24%** | 6.54% | 20.21% | **75.75%** | 5.95% | 18.30% |
| Cigarette butts are harmless to animals and plants | 20.05% | **56.51%** | 23.43% | 19.35% | **59.55%** | 21.10% |
| Cigarette butts are harmless to fish and sea life | 17.05% | **62.57%** | 20.38% | 14.85% | **67.20%** | 17.95% |

The correct answers are in bold.

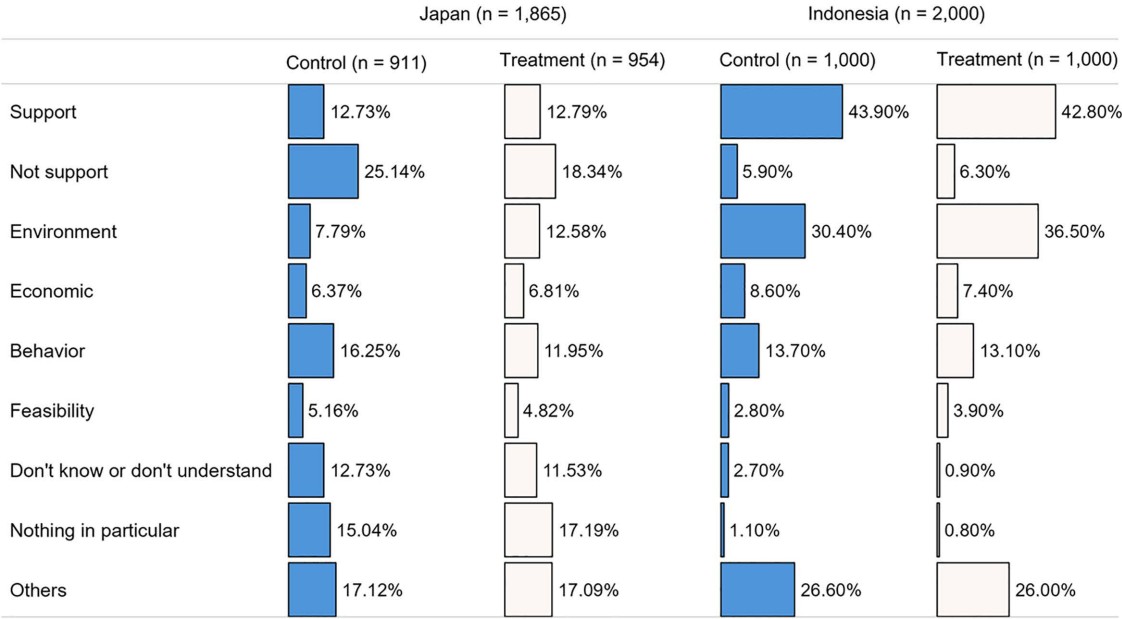

**Fig 5. Respondents' perceptions toward a DRS for CBs.** S1 Supplementary provides complete responses and coding.

(36.50% in Indonesia and 12.59% in Japan). Skepticism about the environmental contributions of a DRS were also raised and categorized under the "environment" theme. Economic-, behavior-, and feasibility-related opinions were generally similar across both countries. The "economic" category included cost-related concerns and views on DRS as an opportunity to generate income. The "behavior" category included smokers' actions and mannerisms, such as unwillingness to return and keep CBs, and stop littering and smoking. The "feasibility" category included perceived challenges or practical considerations regarding the implementation of a DRS. The "others" category included descriptions of explained concepts (e.g., DRS operation, CB recycling), descriptions of their preferences regarding attributes and the respective level, and mentions of DRS in other waste management systems. Irrelevant opinions accounted for 3.84% and 4.93% for the control and treatment groups, respectively, in Japan, and 5.60% and 6.70% for the control and treatment groups, respectively, in Indonesia. We further incorporate respondents' open-ended responses to discuss preferences for RQ3 and RQ4.

### 3.4 Conditional logit model results

Table 4 presents the conditional logit estimates for the control (without environmental information), and the treatment groups (with environmental information) (RQ3, RQ4, RQ5). ASC were positive in both CL1 and CL2 models across countries, implying that respondents preferred the DRS attributes that were not specified in Table 4 to the status quo (i.e., no DRS). The mean coefficients of all the attributes were statistically significant in both countries. While 36.08% and 30.17% of Japanese respondents opted out in the control and treatment groups, respectively, only 9.67% and 8.18% opted out in Indonesia. CL3 models showed that, except for gender in both countries, cigarette consumption in Indonesia, and self-reported littering frequencies in both countries, most smokers' sociodemographic characteristics, behaviors, attitude, and awareness variables significantly influenced DRS preference. Gender's insignificance is attributable to the small sample size of female respondents (Table 2).

Using CL1 and CL2 models, we simulated how smokers' participation probabilities in the DRS changes with varying costs (Fig 6). Cost represents the monetary burden of using the DRS to support its operation. Participation probabilities

**Table 4. Conditional logit estimates.**

| | Japan (n = 1,865) | | | Indonesia (n = 2,000) | | |
|---|---|---|---|---|---|---|
| | CL 1 | CL 2 | CL 3 | CL 1 | CL 2 | CL 3 |
| | Control (n = 911) | Treatment (n = 954) | Control (n = 911) | Control (n = 1,000) | Treatment (n = 1,000) | Control (n = 1,000) |
| Mean coefficients | | | | | | |
| ASC | 0.274*** | 0.481*** | −2.742*** | 1.798*** | 2.077*** | −0.855 |
| | (0.045) | (0.044) | (0.253) | (0.056) | (0.059) | (0.479) |
| Cost | −0.004*** | −0.002*** | −0.004*** | −0.000*** | −0.000*** | −0.000*** |
| | (0.001) | (0.001) | (0.001) | (0.000) | (0.000) | (0.000) |
| Management institution | 0.066** | 0.098*** | 0.070*** | 0.054** | 0.075*** | 0.055** |
| | (0.021) | (0.020) | (0.021) | (0.019) | (0.019) | (0.019) |
| Accessibility | −0.021*** | −0.020*** | −0.022*** | −0.011*** | −0.018*** | −0.011*** |
| | (0.002) | (0.002) | (0.002) | (0.002) | (0.002) | (0.002) |
| ASC:gender | | | −0.056 | | | 0.302 |
| | | | (0.035) | | | (0.159) |
| ASC:age | | | −0.015*** | | | −0.013*** |
| | | | (0.002) | | | (0.004) |
| ASC:education | | | 0.129*** | | | 0.161*** |
| | | | (0.030) | | | (0.046) |
| ASC:income | | | 0.051*** | | | 0.104*** |
| | | | (0.010) | | | (0.018) |
| ASC:cigarette consumption | | | −0.014*** | | | −0.010 |
| | | | (0.004) | | | (0.007) |
| ASC:littering | | | −0.029 | | | 0.015 |
| | | | (0.037) | | | (0.049) |
| ASC:littering inferred | | | 0.291*** | | | −0.150* |
| | | | (0.036) | | | (0.072) |
| ASC:eco-guilt | | | 0.468*** | | | 0.522*** |
| | | | (0.033) | | | (0.065) |
| ASC:knowledge | | | 0.304*** | | | 0.284*** |
| | | | (0.039) | | | (0.064) |
| Opt-out probabilities | 0.3608 | 0.3017 | 0.3608 | 0.0967 | 0.0818 | |
| Adjusted R² | 0.013 | 0.011 | 0.058 | 0.146 | 0.171 | 0.164 |
| AIC | 11859.04 | 12436.99 | 11316.34 | 11253.61 | 10925.29 | 11025.65 |
| BIC | 11885.47 | 12463.6 | 11402.22 | 11280.41 | 10952.09 | 11112.75 |
| Log-likelihood | −5925.521 | −6214.494 | −5645.168 | −5622.804 | −5458.645 | −5499.827 |
| Num. events | 5466 | 5724 | 5466 | 6000 | 6000 | 6000 |

***p < 0.001; **p < 0.01; *p < 0.05. Standard errors are shown in parentheses. ASC, alternative-specific constant; AIC, Akaike information criterion; BIC, Bayesian information criterion.

were generally lower in Japan, and declined more steeply as costs increased. At a cost equal to 10% of the cigarette pack price presented in the DCE choice sets (50 JPY of 500 JPY in Japan and 3,000 IDR of 32,000 IDR in Indonesia), participation probabilities were approximately 46% in Japan and 78% in Indonesia. Treatment groups showed higher participation probabilities, rising to 55% in Japan and 82% in Indonesia at the same cost level.

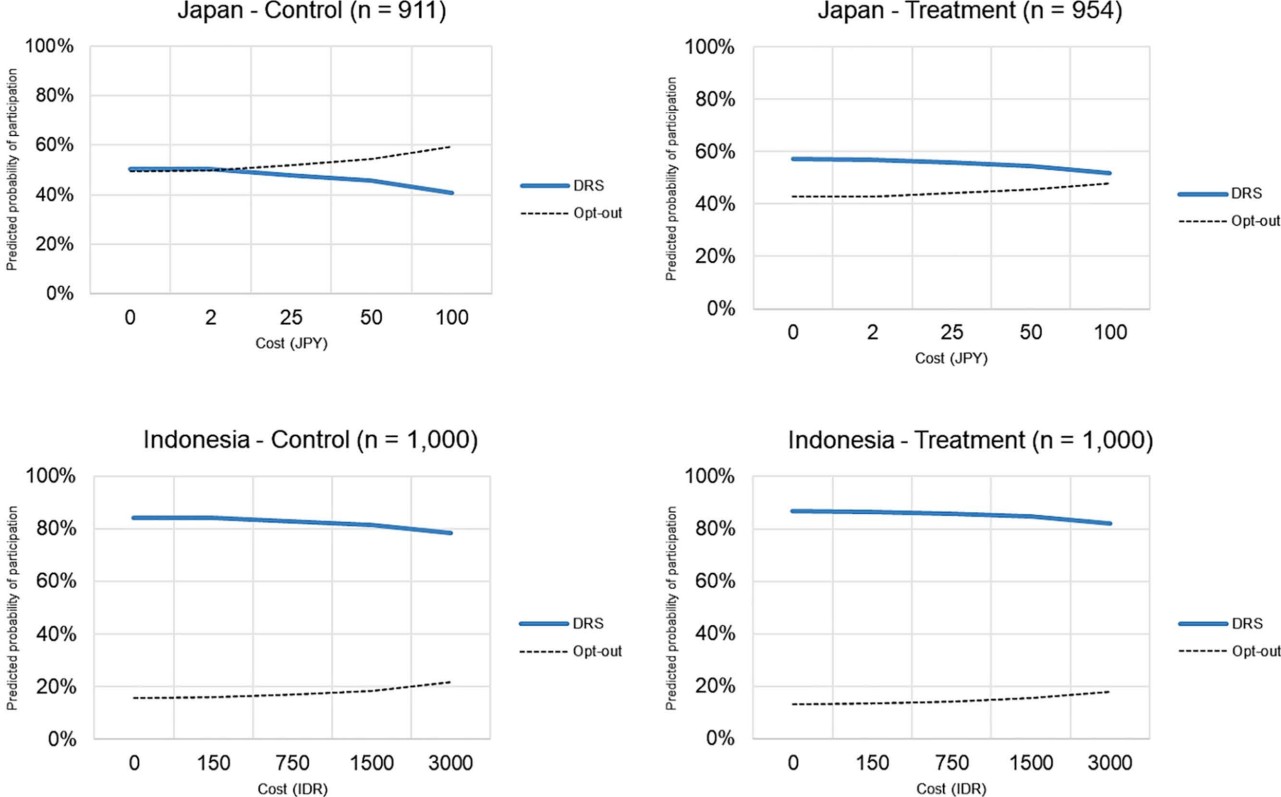

**Fig 6. Simulated adoption rate by cost.** Opt-out refers to respondents who preferred the status quo (i.e., no DRS). They are conditional probabilities tied to the logit model structure.

The WTP tendencies (i.e., the order of preferences over levels and the impact of information on them) were similar between Japanese and Indonesian respondents, as shown in Fig 7. WTP was higher for producer-managed DRS than for government-managed DRS, with the gap widening in the treatment groups. Japanese respondents exhibited a higher WTP for producer-managed DRS than Indonesian respondents (Fig 8). WTP for accessibility was negative, with a marginal decrease of −5.48 JPY for Japan and −84.42 IDR for Indonesia per additional minute required to reach return points. This worsened in the treatment groups. A greater decline was observed among Japanese respondents than Indonesian respondents (Fig 8).

### 3.5 Heat-not-burn cigarettes

HNB cigarette users were more prevalent among Japanese than Indonesian respondents, as was familiarity with the products (77.48% and 51.10%, respectively, Fig 9). Accordingly, Japanese respondents showed significantly higher DRS preferences for HNB cigarettes than Indonesian respondents in both the control and treatment groups (Fig 10, RQ3, RQ4). While Indonesian respondents in the control group exhibited a significantly higher preference for DRS for CBs, no statistically significant difference was observed in the treatment group.

## 4. Discussion

To propose an effective DRS design for CBs, we investigated smokers' understanding of CB's environmental impact (RQ1), perceptions of DRS (RQ2), preference for DRS (RQ3), impact of environmental information on DRS preference (RQ4), and influence of smokers' characteristics on DRS preference over the status quo (RQ5).

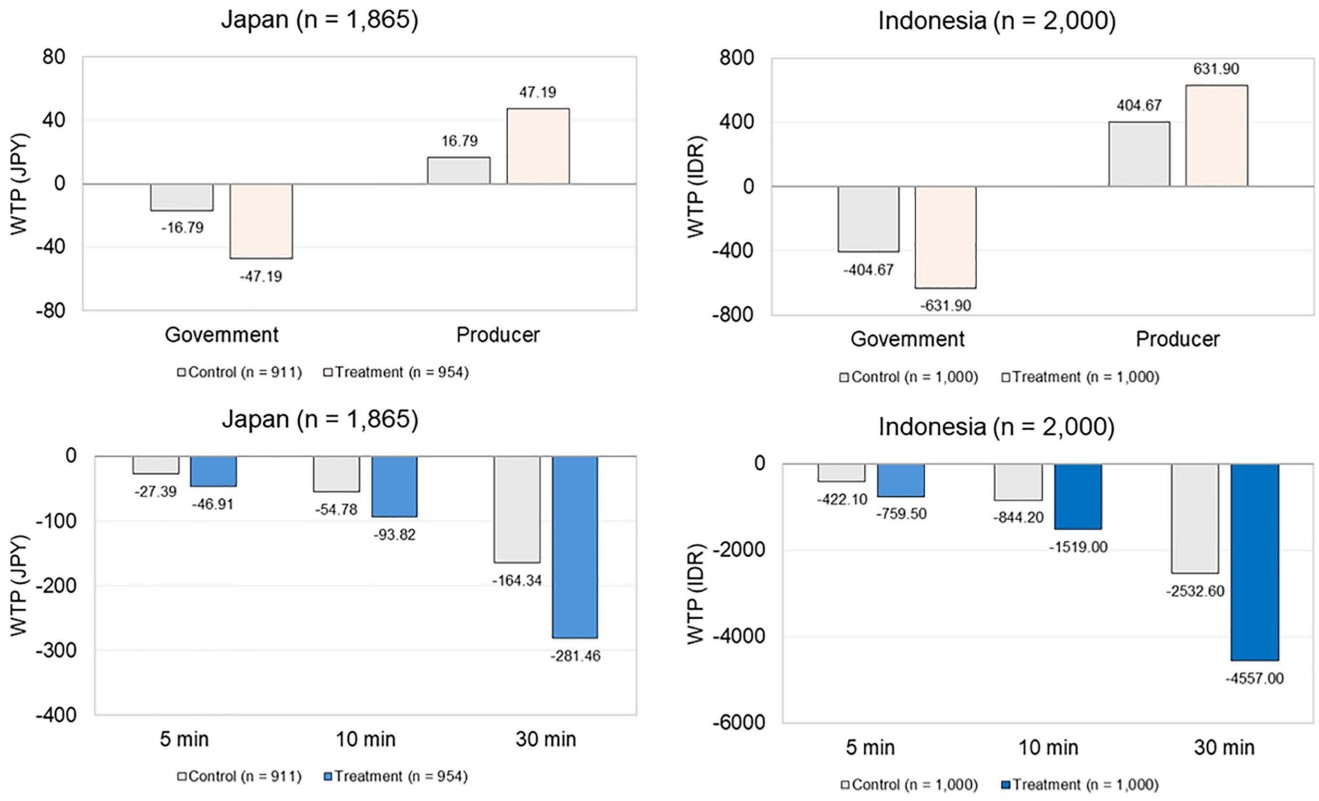

**Fig 7. WTP monetary comparison.** S5 Appendix provides detailed means and standard errors.

### 4.1 Smokers' understanding of the environmental impact of CBs [RQ1]

Japanese and Indonesian respondents in our study demonstrated a better understanding of the environmental impact of CBs than those in previous studies [35,37,90], except for non-biodegradability (Table 3). Approximately 61.45% and 67.60% of Japanese and Indonesian respondents, respectively, either believed that filters are biodegradable or were unsure of their biodegradability. In contrast, a previous study in the United States found that 70% of smokers correctly identified filters as non-biodegradable [38]. The non-biodegradability of CBs was surprising to several Indonesian FG participants, who assumed that CBs would naturally break down over time. This is ascribable to limited knowledge of synthetic materials in cigarette filters [36,91]. Some Indonesian respondents also mentioned their use of kretek, a type of Indonesian cigarettes often sold unfiltered, which may have hindered their understanding of regular filtered cigarettes (responses No. ID 284, ID 731).

### 4.2 Perception of DRS [RQ2]

Inductive coding of DRS perceptions (Fig 5) revealed that Japanese respondents were less supportive of DRS for CBs, than their Indonesian counterparts. This was expected, as Japanese respondents had higher opt-out probabilities (Table 4). Their lower support primarily stemmed from the perception that DRS for CBs would be cumbersome (No. JP 374, JP 402, JP 503), despite some having prior experience with similar systems for beverage containers (No. JP 19, JP 69, JP 339). First, respondents associated the burden with CB storage. For example, one respondent stated, "It seems tough to have to collect and store cigarette butts (No. JP 147)." Unlike beverage containers, CBs are perceived

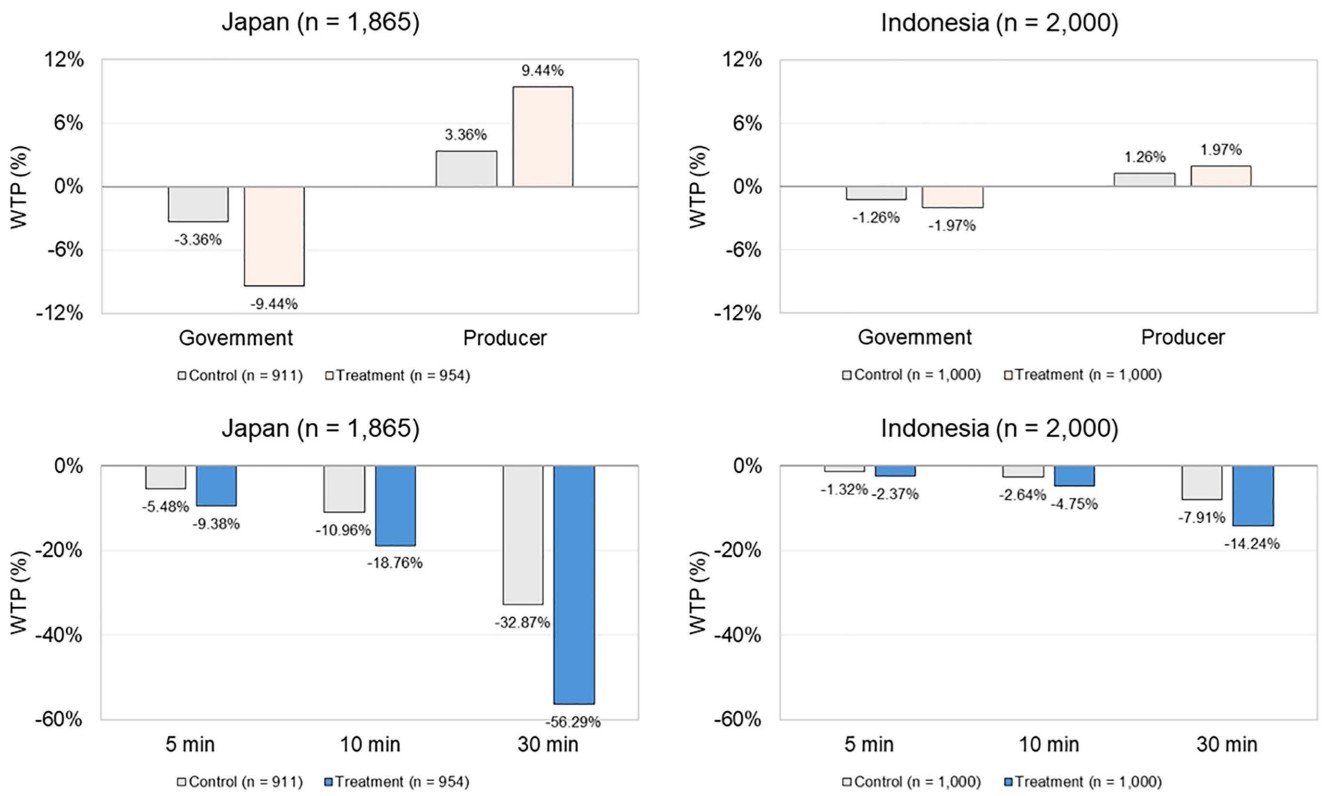

**Fig 8. WTP percentage comparison.**

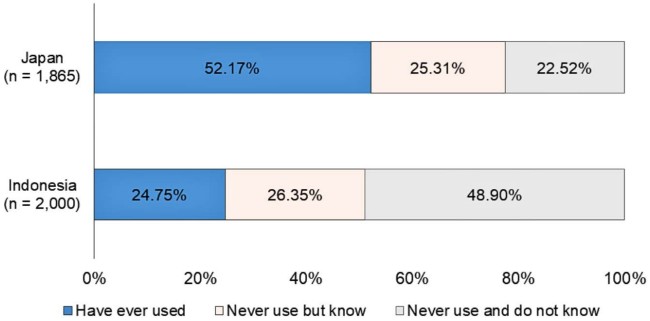

**Fig 9. HNB cigarette usage and familiarity.**

as dirty and smelly [24], and cannot be washed off (No. JP 92). This confirms how object characteristics influence consumer behavior in DRS [59]. Second, respondents linked inconvenience to home smoking and proper disposal practices. One respondent noted, "I think it is effective for people who litter, but since I only smoke at home and always dispose of everything properly, having to take my cigarette butts elsewhere would be quite a burden (No. JP 1301)." Lower support among Japanese respondents also stemmed from the belief that alternative measures, such as expanding smoking areas (No. JP 116, JP 531, JP 747) or carrying portable ashtrays (No. JP 490), would be more effective in reducing CB litter. Additionally, some respondents felt that other environmental issues should take precedence over

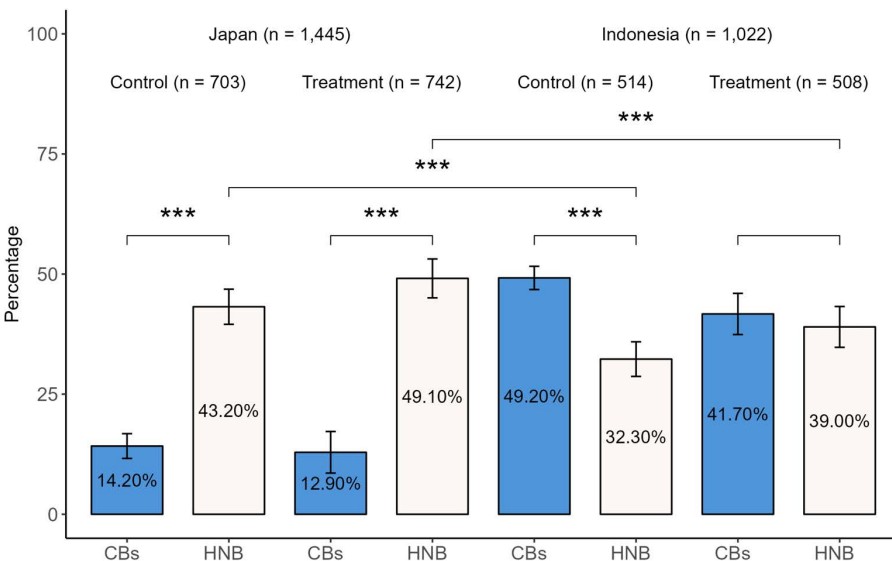

**Fig 10. DRS preferences across tobacco products.** Welch's t-test, consisting of those who have ever use or never use but know HNB cigarettes from Fig 9. ***p < 0.001; **p < 0.01; *p < 0.05. Error bars indicate 95% confidence interval. S6 Appendix provides the complete t-test results.

CB management (No. JP 454). By contrast, higher support among Indonesian respondents was largely associated with environmental concerns. They believed that such a system would benefit the environment (No. ID 10, ID 28, ID 202), a factor that influences positive attitudes toward DRS, including perceptions of it as good, useful, and satisfying [68].

Indonesian respondents expressed more environmentally focused opinions than Japanese respondents. This may be explained by the findings of Kaplan Mintz et al. [92], who indicated that Japanese participants were less likely to be environmentally oriented in waste management than participants from other countries.

## 4.3 Preference for DRS [RQ3]

Both Japanese and Indonesian respondents preferred a mandatory DRS for CBs in the absence of such a system. Notably, preferences were higher in Indonesia, where 90.33% of respondents did not opt out, compared with 63.92% in Japan (Table 4). This suggests that respondents recognized the effectiveness of DRS in reducing CB littering, with their preferences exceeding 51% of smokers in New Zealand, who viewed cigarette deposits as an effective anti-littering measure [35]. Respondents remarked, "I think it would be a wonderful system if this system eliminates littering (No. JP 156)" and "The concept is unique and good. It can help minimize cigarette butts being discarded carelessly (No. ID 22)".

Respondents in both countries preferred a lower cost for DRSs for CBs (CL 1 model, Table 4). This suggests a reluctance to bear additional financial burdens to support the system's operation. This may reflect concerns about already high tobacco taxes, and a desire to avoid further costs (No. JP 146, JP 287, ID 296, ID 466). Compared with Japanese respondents, Indonesian respondents were more willing to endure cost, which is expected, given their generally higher preference and support (Fig. 5). For Indonesian respondents, a cost at or below 5% of the cigarette price (1,500 IDR of 32,000 IDR) maintained participation above 80%. In contrast, in Japan, only costs at or below 2.5% of the cigarette price (2.5 JPY of 500 JPY) maintained participation above 50%.

Respondents in both countries preferred producer-managed DRS over government-managed DRS (CL 1 model, Table 4). This preference is likely driven by the belief that producers should be responsible for CB waste (No. ID 28, ID

491, ID 501). Similarly, Hoek et al. [35] found that 40.9% of New Zealand smokers believed that producers should be responsible for tobacco waste, while 28.3% favored government responsibility. Other reasons cited included the perceived feasibility of producer-managed DRS for CBs (No. ID 434, ID 822), and the desire to minimize the use of public tax funds (JP FG, No. JP 152). This preference aligns with the WHO's [3] recommendations under the EPR framework for managing tobacco-related waste. When comparing the two countries, Japanese respondents exhibited a higher WTP for producer-managed DRS for CBs, than Indonesian respondents (Fig 8). This is attributable to their familiarity with existing producer-led collection programs in Japan for tobacco products such as HNB products, e-cigarettes, and nicotine pouches [31], which are not currently available in Indonesia.

Both Japanese and Indonesian respondents emphasized DRS accessibility (CL 1 model, Table 4). This aligns with consumers' tendency to prefer the nearest facility when travel is required [93], and mirrors the findings on beverage containers DRS [94,95]. Compared with Indonesian respondents, Japanese respondents stressed greater accessibility constraints (Fig 8). This is attributable to their lower support and preference for DRS for CBs, making them less willing to travel to return CBs. Several Indonesian FG participants noted that accessibility was not a concern as cigarette availability implied that return points would also be widespread, which may explain their greater tolerance.

Interestingly, Japanese and Indonesian respondents exhibited opposing preferences when asked to choose between DRS for CBs or for HNB cigarettes (Fig 10). Japanese respondents preferred HNB to CBs, whereas Indonesian respondents preferred CBs to HNB. This difference may primarily stem from Japanese respondents' greater familiarity with HNB products, with a higher usage rate than that of Indonesian respondents (Fig 9). Another possible reason is the existence of Japan's collection programs for HNB devices, batteries, cartridges, and capsules [30,32], which may be perceived as convenient for returning used HNB cigarettes alongside other components. Tasaki et al. [59] noted that the ability to return items together influences consumer behavior in DRS.

### 4.4 Impact of environmental information on DRS preference [RQ4]

Four important differences were observed between the treatment and control groups in both countries. First, respondents in both countries exhibited a higher preference for DRS for CBs, as reflected in higher non-opt-out probabilities (69.83% in Japan and 91.82% in Indonesia, Table 4). This higher preference may have been driven by eco-guilt. A previous study noted that information interventions influence those with stronger environmental values, but insignificantly affect individuals with lower environmental values [96].

Second, the simulated adoption rates indicated a greater willingness to endure costs to support the system's operation (Fig 6). For Indonesian respondents, a cost at or below 10% of the cigarette price (3,000 IDR of 32,000 IDR) could maintain participation above 80%. In contrast, in Japan, costs at or below 5% of the cigarette price (25 JPY of 500 JPY) could maintain participation above 55%. This may suggest a shift in smokers' motivation from avoiding financial burden to contributing to environmental protection. As one respondent stated, "50% refund is beneficial for the earth (No. ID 1078)."

Third, respondents in both countries exhibited higher preferences (CL 2 model, Table 4) and WTP (Fig 8) for producer-managed than government-managed DRS. This increase may be explained by both groups correcting their understanding of CBs' non-biodegradability (Table 3) or gaining a better understanding of their overall environmental impact, which consequently reinforced their perception of producer responsibility for CB waste. This aligns with Hoek et al. [35], who found that smokers' perception of producer responsibility for tobacco waste increased when they were given facts about CBs.

Fourth, respondents in both countries valued accessibility even more (Fig 8). A possible explanation is that environmental information may have heightened their sense of responsibility to return CBs, rather than abandoning or misusing them, which consequently increased the demand for more accessible return points. They considered accessibility a necessary adjustment to accommodate new behaviors.

 

## 4.5 Influence of smokers' characteristics on DRS preference over the status quo [RQ5]

Our findings confirm that certain characteristics place greater value on DRS than on maintaining the status quo [97]. Younger respondents were more likely to prefer DRS for CBs, similar to previous findings on DRS for CBs [35] and DRS for water sachet litter [58]. Kremel [98] found that young adults are more inclined to participate in DRS because they consider it important to the environment. This is likely because younger people are more likely to face environmental risks in the future [99]. Higher education is consistent with a study on DRS for beverage containers [50], yet differed from findings on DRS for CBs, which showed no consistent difference [35], and DRS for beverage containers, where education was insignificant [58,94]. Generally, respondents with higher education levels are more likely to understand environmental impact and engage in pro-environmental behaviors [100,101]. Respondents with a higher annual income preferred the system, possibly because of the lack of financial burden associated with not returning CBs [58].

Among Japanese respondents, daily cigarette consumption was negatively associated with DRS preference, likely owing to the perceived additional effort and added responsibility. Some indicated that they might quit smoking instead of smoking casually (No. JP 86, JP 461, JP 982). Self-reported littering frequencies were insignificant across countries, whereas the inferred valuations of other smokers' littering frequencies were significant. This discrepancy could be explained by moral considerations and social expectations influencing individuals' self-reports, leading to underreporting of their own actions [60]. Hence, inferred valuations were considered in relation to actual littering behavior. Interestingly, Japanese respondents with a higher inferred valuation of littering (i.e., littering more) were more likely to prefer DRS for CBs, possibly viewing DRS as a designated disposal point to change their littering habits. In contrast, it was a lower inferred valuation of littering (i.e., littering less) for Indonesian respondents, possibly because they preferred their current littering habits and were less willing to return CBs.

Higher levels of eco-guilt have been shown to positively influence pro-environmental behavior [62,64]. Greater knowledge of the environmental impact of CBs confirms that perceptions of waste issues influence consumer behavior in DRS [59]. It has also been linked to pro-environmental behavior [102,103].

## 4.6 Policy implications for an effective DRS

Our study demonstrates that environmental information is a promising tool for increasing smokers' preference for DRS for CBs over the status quo, and enhancing their participation probabilities across cost levels. Nonetheless, current WHO environmental campaigns on tobacco's impact, including CBs, primarily target the general public, policymakers, NGOs, tobacco farmers, and academia [3,4] rather than directly engaging smokers. Our findings evidenced that directly targeting smokers is crucial, as there is room for improving their understanding of the environmental impact of CBs, which is also consistent with previous studies [35,38,90]. Such information is particularly important in countries where smokers may not be environmentally oriented toward DRS. Because the target audience is smokers, the WHO's current information dissemination through online articles, posters, and videos could be enhanced by including advertising campaigns and cigarette packaging labels. A previous study found that 72% and 55% of smokers supported these measures to help them understand environmental impact [35]. Emphasizing the non-biodegradability of CBs is strongly recommended, as we found limited awareness of this issue. Future DRS implementations could also incorporate visual elements into return machines, similar to beverage return machines that feature green illustrations and environmental hashtags [104], to highlight the impact of CBs.

The DRS design should align with the smokers' preferences and characteristics in each country, as our study reveals similarities and differences between Japanese and Indonesian smokers. First, a lower cost was generally preferred, although acceptable levels and participation probabilities varied by country. In countries with higher preference and support, a cost at or below 10% of the cigarette price appears suitable for maximizing participation. In contrast, in countries with lower preference and support, a cost of 5% or below may be more appropriate. A producer-managed DRS was

generally preferred over a government-managed DRS. This presents an opportunity for tobacco producers to fulfill their EPR obligations in reducing CB litter, cleaning up waste, and ensuring safe disposal, as recommended by the WHO [3]. Trust and cooperation from consumers are key to successful EPR implementation [105], and smoker support was evident in this study. When domestic producers are unavailable, retail firms having contracts with importers and manufacturers can take responsibility [106]. Third, accessibility needs may vary across countries. In regions where smokers are more sensitive to accessibility, return points should emphasize convenience. Policymakers could consider locating return points where cigarettes are sold, as suggested by a FG participant in our study. This could minimize the time and effort required for return [59]. Fourth, DRS should consider the familiarity and usage of different tobacco products in each country. In regions where HNB cigarettes are widely used, DRS should prioritize their collection alongside CBs. Related studies showed that HNB cigarettes pose environmental toxicity risks [107,108], making their collection crucial for reducing the environmental impact. Existing collection systems, such as those in Japan, use retailers to collect devices [30–32]; these systems could be expanded to include cigarette sticks as well. Fifth, targeted outreach efforts should prioritize individuals who prefer the status quo and resist behavioral change, such as those who are older, have lower educational and income levels, exhibit lower eco-guilt, and have limited knowledge of CBs. In countries where smokers are unwilling to change their littering habits and unlikely to return CBs, handling portable ashtrays could be a potential solution, as suggested by a FG participant. These would not only help manage CB dirt and odor, but also prevent littering [13,16]. By making it easier to carry CBs to return points, portable ashtrays may positively influence DRS preference.

## 5. Conclusion

This paper is the first to examine smokers' perceptions and preferences regarding a DRS for CBs. Using a stated preference DCE, we explored a range of scenarios, attributes, levels, and information interventions that would be difficult to replicate in a real-world setting, thereby contributing to the literature on effective DRS design. Our study revealed the importance of how smokers perceive deposit and refund rate as a combined attribute, which we defined as cost, an approach that can inform and benefit future DRS research. Additionally, we attempted to simulate adoption rates based on this cost measure, extending prior research by offering insights into how varying cost levels may influence participation probabilities and inform policy decisions aimed at maximizing uptake. We found that although perceptions of DRS for CBs varied, smokers generally preferred it to the status quo. The treatment-control design revealed that environmental information increased smokers' preferences for DRS for CBs to the status quo, and induced higher participation probability across cost levels. This highlights the importance of integrating environmental information into DRS design, and suggesting that WHO's current campaign should directly target smokers. Smokers' awareness of the environmental impact of CBs could be enhanced, particularly regarding their non-biodegradability. The DRS design should be tailored to each country as some preferences and characteristics vary contextually. For instance, while a producer-managed DRS was generally preferred to a government-managed DRS, differences emerged in the acceptability of cost and accessibility, tobacco product choices, and individual littering behavior that influences DRS preference. Furthermore, efforts to target individuals who prefer the status quo should be made.

This study has three limitations. First, because we employed a stated preference methodology, the responses may not fully capture actual preferences. Moreover, the hypothetical scenarios do not consider smokers' real-world behavior in returning CBs. The simulated adoption rates by cost were conditional and may not correspond to actual participation rates. Choice experiments are often subject to hypothetical bias and require external validation [109,110]. Thus, a field experiment could help validate our findings. Although implementing DRS for CBs may be technically challenging, it is not necessarily impossible given current technologies. Recreational beaches could serve as ideal sites for such experiments due to their consistently high concentration of CBs and associated metal toxicity leaching [15], as well as the precedent set by behavioral nudge interventions involving smokers [16]. Second, further research on DRS for HNB cigarettes is required. As HNB products become more widely adopted, understanding smokers' preferences is crucial. Although we

attempted to assess these preferences, we did not examine how these preferences changed based on attributes, levels, or exposure to environmental information. Applying a similar DCE design to HNB cigarettes would clarify the similarities and differences in DRS designs across tobacco products. Third, underage smokers were not included, which may have limited the findings' generalizability. Future studies could address this limitation by considering underage smokers where legally and ethically appropriate, as their numbers are not negligible in some countries.

## Supporting information

**S1 Appendix.  Eco-guilt items.**
(DOCX)

**S2 Appendix.  DRS scenario (Japan*).** * Indonesia only differs in the assumption that one pack of cigarettes (16 sticks) is priced at IDR 32,000 or IDR 2,000 per stick.
(DOCX)

**S3 Appendix.  Conditional logit estimates using deposit and refund rate attributes.**
(DOCX)

**S4 Appendix.  Welch's test for behavior and attitude characteristics of respondents (Fig 4).**
(DOCX)

**S5 Appendix.  WTP means and standard errors (Fig 7).** We adopted Krinsky and Robb's method to calculate 95% confidence intervals [48]. The WTP for accessibility appears different in Fig 7 because it has been multiplied to reflect the respective attribute levels.
(DOCX)

**S6 Appendix.  Welch's test for DRS preferences for tobacco products (Fig 10).**
(DOCX)

**S1 Supplementary.  Data.**
(XLSX)

**S2 Supplementary.  Sample questionnaire.**
(DOCX)

## Author contributions

**Conceptualization:** Fivita Stri.

**Data curation:** Fivita Stri, Takuro Uehara.

**Formal analysis:** Fivita Stri, Takuro Uehara, Takahiro Tsuge, Sitadhira Prima Citta.

**Funding acquisition:** Takuro Uehara, Misuzu Asari.

**Investigation:** Fivita Stri, Takuro Uehara.

**Methodology:** Fivita Stri, Takuro Uehara, Takahiro Tsuge.

**Project administration:** Takuro Uehara.

**Resources:** Fivita Stri, Takuro Uehara.

**Software:** Fivita Stri.

**Supervision:** Takuro Uehara.

**Validation:** Fivita Stri, Takahiro Tsuge.

**Visualization:** Fivita Stri.

**Writing – original draft:** Fivita Stri.

**Writing – review & editing:** Takuro Uehara, Takahiro Tsuge, Sitadhira Prima Citta, Misuzu Asari.

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
