## [Decision Letter · Decision Letter 0]

22 Aug 2025

PONE-D-25-33751Designing a deposit-refund system for cigarette butts: What do smokers care about?PLOS ONE

Dear Dr. Uehara,

Thank you for submitting your manuscript to PLOS ONE. After careful consideration, we feel that it has merit but does not fully meet PLOS ONE’s publication criteria as it currently stands. Therefore, we invite you to submit a revised version of the manuscript that addresses the points raised during the review process.

We look forward to receiving your revised manuscript.

Kind regards,

Chih-Cheng Lin, Ph.D.

Academic Editor

PLOS ONE

**Comments to the Author**

1. Is the manuscript technically sound, and do the data support the conclusions?

Reviewer #1: Partly

Reviewer #2: Yes

2. Has the statistical analysis been performed appropriately and rigorously? 

Reviewer #1: Yes

Reviewer #2: Yes

3. Have the authors made all data underlying the findings in their manuscript fully available?

Reviewer #1: No

Reviewer #2: Yes

4. Is the manuscript presented in an intelligible fashion and written in standard English?

Reviewer #1: Yes

Reviewer #2: Yes

5. Review Comments to the Author

Reviewer #1: This manuscript presents a well-executed discrete choice experiment (DCE) exploring smokers’ preferences for a deposit-refund system (DRS) for cigarette butts in Japan and Indonesia. It contributes to the literature on extended producer responsibility and behavioural environmental policy by incorporating both stated preferences and an information intervention. The paper is well-organised and clearly written, with sound methodology. However, there are several conceptual and interpretive issues that require attention before publication.

Main comments

1. Scope and Framing of Research Questions

The manuscript presents five research questions, but the mapping between questions, methods, and results is not always clear. For instance, the link between RQ2 (perceptions) and the inductive coding of open-ended responses is weaker than the structured linkages for RQ3–RQ5. I recommend that the authors:

• Clarify which analytical components correspond to each research question.

• Structure the results section to more clearly reflect this mapping.

2. DCE Design vs. Claims About Feasibility

The introduction and abstract suggest that the study addresses the general feasibility of implementing a DRS for tobacco products. However, the DCE design only captures stated preferences for attributes of a hypothetical scheme. This is not equivalent to assessing real-world feasibility or predicting actual adoption or compliance. The authors should revise claims in the abstract, introduction, and conclusions to focus on preferences rather than feasibility per se.

3. Limitations of DCEs in the Tobacco Context

Stated preference methods, such as DCEs, face well-documented challenges in contexts like smoking behaviour, which involve automaticity, time-inconsistent preferences, and attentional limits. These characteristics reduce the external validity of DCEs in predicting actual behaviour.

The discussion and conclusion should more explicitly acknowledge these behavioural and methodological limitations, drawing on the following literature:

• Regmi et al. (2018): A systematic review of DCEs in tobacco control highlights limited behavioural realism and strong influence of monetary attributes in stated preferences.

Regmi, K., Kaphle, D., Timilsina, S., & Tuha, N. A. A. (2018). Application of discrete-choice experiment methods in tobacco control: a systematic review. PharmacoEconomics-Open, 2(1), 5–17.

https://doi.org/10.1007/s41669-017-0030-9

• Quaife et al. (2018): This meta-analysis finds that DCEs have only modest external validity, with an area under the curve (AUC) of 0.68, raising questions about their predictive power for real-world choices.

Quaife, M., Terris-Prestholt, F., Di Tanna, G. L., & Vickerman, P. (2018). How well do discrete choice experiments predict health choices? A systematic review and meta-analysis of external validity. The European Journal of Health Economics, 19(8), 1053–1066.

https://doi.org/10.1007/s10198-018-0960-1

• Haghani et al. (2021): This large-scale review identifies consistent hypothetical bias in stated choice experiments, especially in behavioural domains involving low attention or habitual actions, such as tobacco consumption.

Haghani, M., Bliemer, M. C., Rose, J. M., Oppewal, H., & Lancsar, E. (2021). Hypothetical bias in stated choice experiments: Part I. Macro-scale analysis of literature and integrative synthesis of empirical evidence from applied economics, experimental psychology and neuroimaging. Journal of Choice Modelling, 41, 100309.

https://doi.org/10.1016/j.jocm.2021.100309

This discussion should be used to limit the interpretation of findings and clearly state that the DCE results reflect hypothetical preferences, not observed behaviour or implementation success.

Recommendations on Tables, Figures, and Presentation

1. Figures 5–7 (WTP and Participation Rates):

o Explicitly explain in the text how WTP values are derived from model coefficients.

o Consider providing a table with means and standard errors for WTP comparisons to support inferences from Figure 6.

o Clarify in the figure or caption whether participation rates are conditional or absolute.

2. Table 4 (Conditional Logit Estimates):

o Add statistical significance indicators (e.g., asterisks) for interaction terms.

3. Table 1 (Thematic Coding of Perceptions):

o Refer to Table 1 more directly when discussing open-ended perceptions in Section 4.2.

o Clarify whether themes map back to RQ2 explicitly.

Minor Comments

• Correct the typo in the abstract: "90.33%%" → "90.33%".

• Consider defining “heat-not-burn cigarettes” upon first mention for readers unfamiliar with the term.

• Confirm that all country-specific monetary values and exchange rates used in the WTP conversion are properly footnoted or sourced.

Reviewer #2: Strengths:

The study addresses a significant environmental issue (cigarette butt pollution) using a novel approach (deposit-refund system, DRS).

The comparative design (Japan vs. Indonesia) provides valuable insights into cultural and socioeconomic differences in smokers' perceptions and preferences.

The use of a discrete choice experiment (DCE) and treatment-control design is methodologically robust and appropriate for assessing hypothetical scenarios.

The inclusion of environmental information intervention is a strong point, as it highlights the role of awareness in shaping behavior.

Areas for Improvement:

Introduction: The introduction could better contextualize the problem of cigarette butt pollution by citing recent studies, such as the comparative study on recreational beaches (referenced below). This would strengthen the justification for the study.

Sampling Methods: The manuscript could benefit from a discussion of the limitations of self-reported data (e.g., social desirability bias) and how the inferred valuation method mitigates this.

Policy Implications: While the policy implications are well-articulated, they could be expanded to include specific recommendations for integrating DRS with existing waste management systems in each country.

Figures and Tables: Ensure all figures and tables are clearly labeled and referenced in the text. Some figures (e.g., Fig 9) lack descriptive captions.

Minor Issues:

Typos and grammatical errors (e.g., "as follow-up to 'Financial Disclosure'" appears twice redundantly).

The abstract could be more concise; some sentences are overly long.

Suggested Improvements for the Introduction Using the Cited Article

The attached article, "Assessing cigarette butt pollution on recreational beaches: A comparative study of two sampling methods and their impact on metal release," provides empirical evidence of cigarette butt pollution's environmental impact, particularly in marine environments. Here’s how you can integrate its findings into your introduction:

Current Problem:

Add: "Cigarette butts are not only the most littered item globally but also a significant source of toxic metal contamination in coastal ecosystems. Recent studies have demonstrated that cigarette butts leach heavy metals like cadmium, lead, and arsenic into beach environments, posing risks to marine life and human health (cite the beach study). This underscores the urgency of addressing cigarette butt pollution beyond urban settings."

*****use this article in manuscript to improve introduction "Assessing cigarette butt pollution on recreational beaches: A comparative study of two sampling methods and their impact on metal release:

Gaps in Existing Measures:

Expand: "While measures like fines and clean-up initiatives target urban littering, they are less effective in recreational areas such as beaches, where cigarette butts are often buried in sand and overlooked by traditional sampling methods (cite the beach study). This highlights the need for systemic solutions like DRS, which can incentivize proper disposal across diverse environments."

Justification for DRS:

Link: "The persistence of cigarette butts in marine environments, as evidenced by their high metal leaching potential (cite the beach study), aligns with our findings on smokers' misconceptions about biodegradability. A DRS could mitigate this by ensuring proper collection and recycling, reducing environmental toxicity."

Comparative Context:

Emphasize: "The beach study's comparison of sampling methods also mirrors our cross-country approach, revealing how context-specific factors (e.g., beach tourism in Indonesia vs. urban littering in Japan) influence pollution patterns. Tailoring DRS designs to these differences is critical for success."

Additional Recommendations

Clarify Terminology: Define "heat-not-burn (HNB) cigarettes" early in the introduction, as not all readers may be familiar with this product.

Highlight Innovation: Stress how your study advances prior work (e.g., by testing cost thresholds and producer-managed systems, which are underexplored in the literature).

Future Research: Suggest field experiments to validate DCE results, citing the beach study's methodological rigor as a model.

6. PLOS authors have the option to publish the peer review history of their article (what does this mean? ). If published, this will include your full peer review and any attached files.

**Do you want your identity to be public for this peer review?** For information about this choice, including consent withdrawal, please see our Privacy Policy .

Reviewer #1: No

Reviewer #2: No

---

## [Author Response · Author response to Decision Letter 1]

2 Sep 2025

Thank you for your comments and suggestions. We have uploaded the response letter as an attachment.

---

## [Decision Letter · Decision Letter 1]

8 Oct 2025

Designing a deposit-refund system for cigarette butts: What do smokers care about?

PONE-D-25-33751R1

Dear Dr. Uehara,

We’re pleased to inform you that your manuscript has been judged scientifically suitable for publication and will be formally accepted for publication once it meets all outstanding technical requirements.

Kind regards,

Chih-Cheng Lin, Ph.D.

Academic Editor

PLOS ONE

**Comments to the Author**

1. If the authors have adequately addressed your comments raised in a previous round of review and you feel that this manuscript is now acceptable for publication, you may indicate that here to bypass the “Comments to the Author” section, enter your conflict of interest statement in the “Confidential to Editor” section, and submit your "Accept" recommendation.

Reviewer #1: All comments have been addressed

Reviewer #2: All comments have been addressed

2. Is the manuscript technically sound, and do the data support the conclusions?

Reviewer #1: Yes

Reviewer #2: Yes

3. Has the statistical analysis been performed appropriately and rigorously? 

Reviewer #1: Yes

Reviewer #2: Yes

4. Have the authors made all data underlying the findings in their manuscript fully available?

Reviewer #1: No

Reviewer #2: Yes

5. Is the manuscript presented in an intelligible fashion and written in standard English?

Reviewer #1: Yes

Reviewer #2: Yes

6. Review Comments to the Author

Reviewer #1: The authors have carefully addressed all the comments raised in the previous round. The response letter was particularly clear and helpful in assessing how each point was handled. I would just recommend that they also upload the dataset and relevant code to an open repository to strengthen transparency and reproducibility.

Reviewer #2: (No Response)

7. PLOS authors have the option to publish the peer review history of their article (what does this mean? ). If published, this will include your full peer review and any attached files.

**Do you want your identity to be public for this peer review?** For information about this choice, including consent withdrawal, please see our Privacy Policy .

Reviewer #1: **Yes: ** Paul Rodriguez Lesmes

Reviewer #2: No

---

## [Editor Report · Acceptance letter]

PONE-D-25-33751R1

PLOS ONE

Dear Dr. Uehara,

I'm pleased to inform you that your manuscript has been deemed suitable for publication in PLOS ONE. Congratulations! Your manuscript is now being handed over to our production team.

Kind regards,

on behalf of

Dr. Chih-Cheng Lin

Academic Editor

PLOS ONE